# Comparative Mitogenomics of True Frogs (Ranidae, Anura), and Its Implications for the Phylogeny and Evolutionary History of *Rana*

**DOI:** 10.3390/ani12101250

**Published:** 2022-05-12

**Authors:** Wan Chen, Weiya Qian, Keer Miao, Ruen Qian, Sijia Yuan, Wei Liu, Jianhua Dai, Chaochao Hu, Qing Chang

**Affiliations:** 1Jiangsu Key Laboratory for Biodiversity and Biotechnology, School of Life Sciences, Nanjing Normal University, Nanjing 210046, China; wanwan0322@163.com (W.C.); qianweiya@126.com (W.Q.); nicolemke@163.com (K.M.); dai-jianhua@163.com (J.D.); 2College of Environment and Ecology, Jiangsu Open University (The City Vocational College of Jiangsu), Nanjing 210036, China; qianruen@163.com (R.Q.); sijiayuanlucas@163.com (S.Y.); 3Nanjing Institute of Environmental Sciences, Ministry of Environmental Protection, Nanjing 210042, China; lwecology@126.com; 4Analytical and Testing Center, Nanjing Normal University, Nanjing 210046, China

**Keywords:** *Rana*, mitochondrial genome, gene order, gene arrangement, phylogeny

## Abstract

**Simple Summary:**

The true frogs of the genus *Rana* are a complex and diverse group. Many new species have been discovered with the help of molecular markers and morphological traits. However, the evolutionary history in *Rana* were not well understood. In this study, we sequenced and annotated the complete mitochondrial genome of *R. longicrus* and *R. zhenhaiensis*. In 13 protein codon genes, the COI was the most conserved, and ATP8 had a fast rate of evolution. The Ka/Ks ratio analysis among *Rana* indicated the protein-coding genes were suffering purify selection. There were three kinds of gene arrangement patterns found. This study provides mitochondrial genetic information, improving our understanding of mitogenomic structure and evolution, and recognizes the phylogenetic relationship and taxonomy among *Rana*.

**Abstract:**

The true frogs of the genus *Rana* are a complex and diverse group, containing approximately 60 species with wide distribution across Eurasia and the Americas. Recently, many new species have been discovered with the help of molecular markers and morphological traits. However, the evolutionary history in *Rana* was not well understood and might be limited by the absence of mitogenome information. In this study, we sequenced and annotated the complete mitochondrial genome of *R. longicrus* and *R. zhenhaiensis*, containing 22 tRNAs, 13 protein-coding genes, two ribosomal RNAs, and a non-coding region, with 17,502 bp and 18,006 bp in length, respectively. In 13 protein codon genes, the COI was the most conserved, and ATP8 had a fast rate of evolution. The Ka/Ks ratio analysis among *Rana* indicated the protein-coding genes were suffering purify selection. There were three kinds of gene arrangement patterns found. The mitochondrial gene arrangement was not related to species diversification, and several independent shifts happened in evolutionary history. Climate fluctuation and environmental change may have played an essential role in species diversification in *Rana*. This study provides mitochondrial genetic information, improving our understanding of mitogenomic structure and evolution, and recognizes the phylogenetic relationship and taxonomy among *Rana*.

## 1. Introduction

The Ranidae (Anura) is comprised of 24 genera and approximately 434 species worldwide [1]. The true frogs of the genus *Rana* are a complex and diverse group that are widely distributed across Eurasia and the Americas [2,3]. Due to their body coloration and habitat preferences, the species in *Rana* are commonly known as brown frogs or wood frogs [4,5,6]. In the genus *Rana*, approximately 60 species are recorded in the world [7], and five of seven clades exist in China [2], including 25 species have been recorded [8]. The conservative morphology of *Rana* makes many species difficult to identify [9]. Moreover, with the rapid and notable developments in the knowledge about the true frogs, many new species have been discovered in China with the help of molecular markers associated with morphological traits in recent years [4,10,11,12]. The recent study of new species descriptions has accelerated quickly, suggesting that insufficiently explored regions may contain many cryptic new species, indicating that the diversity of this genus is probably underestimated [13,14].

The typical mitochondrial genome (mitogenome) of vertebrates is a closed ring structure with a size range of 15–21 kb, which contains 13 protein-coding genes (PCGs), 2 ribosomal RNAs (rRNAs), 22 transfer RNAs (tRNAs), and one non-coding region (D-loop) [15,16]. Due to its small genome size, compact gene arrangement, high conservation, low sequence recombination, maternal inheritance, and easy detection, mitogenome provides a valuable resource for further study of molecular systematics, population genetics, comparative or evolutionary genomics [17,18,19,20]. To date, the advances of next-generation sequencing (NGS) techniques offer new opportunities for rapid increasingly high data quality of mitogenomes [21,22].

Recent studies have discovered many new species in the genus of *Rana* from China and revised several taxonomic arrangements, indicating that the diversity and taxonomy of the *Rana* are still not well understood [11]. True frogs are extensively used as model organisms in studies of development, genetics, physiology, behavior, ecology, and evolution [2]. Despite recent rapid increases in the available information on *Rana* mitogenomes, only 20 complete mitogenomes sequences have been reported in GenBank (accessed on 24 October 2021, Table 1). However, the basic genetics information and characterization of mitogenome of *Rana* are unclear.

In this study, we sequenced and annotated the complete mitogenome of two *Rana* species (*R. longicrus* and *R. zhenhaiensis*). The genome composition and characteristics were analyzed, the relative synonymous codon usage (RSCU) and AT-skew values of the PCGs were calculated. Furthermore, we analyzed phylogenetic relationships, gene arrangement, and selective pressures among *Rana* species. In the future, sequencing more mitogenomes from various taxonomic levels will provide useful information for better understanding the evolutionary history and will contribute to further taxonomic research within Ranidae.

## 2. Materials and Methods

### 2.1. Sample Collection and DNA Extraction

The samples of *R. longicrus* was collected from Chiu-ling Mountains, Jiangxi Province, China (29°12′14.47′′N, 114°52′4.42′′E), and *R. zhenhaiensis* was collected from Tianmu Mountains, Zhejiang Province, China (30°11′0.75″N, 119°27′56.64″E). After collection, the muscles were initially preserved in 75% ethanol for two days, then preserved in 95% ethanol, and finally transferred to −20 °C in the laboratory for long-term storage in Nanjing Normal University. Total genomic DNA was extracted using a DNAeasy tissue kit (Qiagen, Hilden, Germany) following the manufacturer’s instructions.

### 2.2. Library Preparation and Sequencing

The extracted DNA was then sent to Novogene (Beijing, China) for sequencing on the Illumina sequencing platform. The concentration of DNA was checked with a Nanodrop 1000 Spectrophotometer (Thermo Scientific, Waltham, MA, USA). The extracted DNA was sheared to 400–600 bp using an ultrasonic technique. The sequencing library was produced using the Illumina Truseq DNA Sample Preparation Kit (Illumina, San Diego, CA, USA) according to the manufacturer’s instructions. The prepared library was loaded on the Illumina Novaseq 6000 platform for PE 2 × 150 bp sequencing at Novogene (Beijing, China).

A total of 6.0 Gb raw reads were generated by next-generation sequencing on the Illumina platform. Before assembly, Illumina raw data were filtered into clean reads, and then undesirable reads were removed by fastp v. 0.21 [23], with the following parameters “–q 15 –u 40 –5 -–x –w 40 –f 10 –F 10” [23]. This filtering step was performed to remove the reads with adaptors, the reads showing a quality score below 20 (Q < 20), containing a percentage of uncalled based (“N” characters) equal or greater than 10% and the duplicated sequences. After removing low quality sequences, the 5.67 Gb clean Raw Data of *R. zhenhaiensis* and 5.05 Gb of *R. longicrus* were harvested. The clean reads *de novo* assemblies were conducted in Geneious 10.1.2 using the mitogenome of *R**. dybowskii* (GenBank no. KF898355) as a reference map [24], and aligned contigs (≥80% similarity and an average 170 X coverage) were ordered according to the reference genome. The assembled contig identified as mitogenome was manually examined for repeats at the beginning and end of the sequence to infer circularity.

### 2.3. Sequence Assembly and Annotation

The mitogenome was first annotated with MITOS [25]. The PCGs and rRNA genes were determined by NCBI open reading frame finder implemented at the NCBI website with the vertebrate mitochondrial genetic code, and then manually corrected by comparison with the available sequences of closely related *Rana* species downloaded from GenBank using ClustalW in MEGA X [26]. The, tRNAscan-SE [27], and ARWEN [28] were used to confirm tRNA and rRNA annotation results. All tRNA genes were identified by their cloverleaf secondary structure using tRNA-scan SE or determined by comparison with the homologous sequences [27]. The graphical map of the mitogenome map was conducted using CGView Server online software [29].

The software of MEGA X was used to calculate the number of variable sites, the parsimony informative sites, the singleton, and the average uncorrected pairwise distances for 13 protein-coding genes of *Rana* [26]. Codon usage was estimated using DnaSP 5.1 [30]. Nucleotide composition and the relative synonymous codon usage (RSCU) were calculated with MEGA X [26]. The rates of non-synonymous substitutions (Ka, π modified), synonymous substitutions (Ks, π modified), the effective number of codons (ENC) and the codon bias index (CBI) for each protein-coding gene were determined with DnaSP 6.0 [31]. The value of nucleotide composition skewness was measured using the following formulas: GC-skew = (G − C)/(G + C) and AT-skew = (A − T)/(A + T) [32,33]. The tandem repeats were searched in the CR using the Tandem Repeats Finder program (https://tandem.bu.edu/trf/trf.html, accessed on 18 November 2021) [34].

### 2.4. Molecular Phylogenetic Analysis

We constructed the phylogenetic topology of 18 *Rana* species using 13 PCGs, 12S and 16S with two species (*Babina subaspera* GenBank no. NC_02287 and *Pelophylax nigromaculatus* KT878718) as outgroups. Phylogenetic analysis was performed using Bayesian Inference (BI). To determine the optimal partitioning of the data, the best-fit partitioning scheme and the most appropriate nucleotide evolution model for each partition were implemented in PartitionFinder 2, with greedy algorithm and Akaike Information Criterion (AICc) criterion [35]. BI method was performed using MrBayes 3.1.2 [36]. Four Markov Chains Monte Carlo (MCMC) chains were run for 1.0 × 10^6^ generations. Two independent runs were performed to confirm consistent approximation of the posterior parameter distributions. Stationarity was reached when the average standard deviation of split frequencies was below 0.01. The convergence of MCMC runs and effective sample sizes (ESS > 200) were checked by plotting the log-likelihood scores against the generation times using the program TRACER 1.6 (http://beast.bio.ed.ac.uk/Tracer accessed on 18 November 2021). The first 25% of sampled trees and estimated parameters were discarded as burn-in. The remaining trees were used to calculate consensus tree and Bayesian posterior probabilities.

### 2.5. Divergence Times Estimates

Divergence times were estimated using BEAST 1.8.4 [37]. The following parameters were used: GTR + *I* + *G* substitution model, relaxed log-normal molecular clock model and a Yule process for tree prior. Two calibration points were used for time calibration. The first calibration point, representing the split between Eurasian and American species, was set as 31.2 ± 8.1 Ma [2,38]. The second calibration point was a constraint of 15.0 Ma with 2.5% and 97.5% quantiles of 15.2 and 23.6 Ma for the most recent common ancestor of the *R. catesbeiana* group [2,13,39]. We performed two independent runs with a MCMC chains of 2 × 10^8^ generations, and trees were sampled every 1000 generations. The first 25% of the generations were discarded as burn-in. Convergence of the chains was determined with TRACER ver. 1.7.1 [40], with target ESS values more than 200 for all parameters. The tree information was annotated and visualized by FigTree v. 1.4.4 (http://tree.bio.ed.ac.uk/software/figtree/, accessed on 24 October 2021).

### 2.6. Mitogenomic Rearrangements Analysis

Multiple sequence alignment of the *Rana* mitogenome sequences was performed to verify the gene order of the mitochondrial genome. We reconstructed the ancestral states of gene rearrangements. The rearrangements patterns were categorized into three types (result 3.4). Ancestral discrete characteristics were reconstructed in a time-calibrated MCMC tree using the “ace” function in the “ape” package [41]. The parameters of the models were estimated using the ML method [42]. The final states were plotted in the time-calibrated MCMC tree using the “phytools” package [43].

## 3. Results and Discussion

### 3.1. General Characteristics of the Mitogenome

The complete mitogenome of *R. longicrus* and *R. zhenhaiensis* was 17,502 bp and 18,006 bp in length, respectively. They carried the typical composition, including 37 genes (13 PCGs, two rRNAs, and 22 tRNAs) and the control region (Figure 1). Length differences were primarily the result of variation in control region. The overall base composition of the two *Rana* species in descending order was as follows: 29.8% C, 27.7% T, 27.3% A, and 15.1% G for *R. longicrus*; 29.9% C, 27.6% T, 27.4% A, and 15.0% G for *R. zhenhaiensis*. The average length of mitogenome in *Rana* was 18,401 bp, ranging from 16,061 bp (*R. temporaria*) to 22,255 bp (*R. kunyuensis*). The mitogenomes of *Rana* species are consistently biased towards AT rich, with an A + T% range from 55.7% (*R. pyrenaica*) to 60.6% (*R. kunyuensis*).

### 3.2. Protein-Coding Genes and the Codon Usage

Relative synonymous codon usage (RSCU) values for the 13 PCGs were summarized in Table 2. The 13 most frequently used amino acid was Ile^AUU^, which accounts for 4.56% of the usage, and the least was Arg^CGG^ (0.17%). To further study the codon usage bias of *Rana*, the correlation between ENC (effective codon number), CBI (codon bias index), G + C content of all codons (G + Cc), and G + C content of the third codon position (G + C3s) were analyzed. We found a significant negative correlation between CBI and ENC (*R*^2^ = 0.71; *p* < 0.01). However, there was no correlation between the other pairs (Figure 2).

The total length of the protein-coding genes in each species was 11,211 bp after removing termination codons and indels. The length of 13 PCGs ranged from 159 bp (ATP8) to 1740 bp (ND5). The overall AT-skew and GC-skew of the 13 PCGs were negative except that the AT-skew of COII and ATP8 and the GC-skew of ND6 were positive (Table 3). The use of start codons for 13 PCGs were quite common. Four start codons (ATG, GTG, ATC and ATA) were detected in 13 PCGs. The most common start codon was ATG, which accounts for 70.94% of the start codons, followed by GTG (16.67%) (Figure 3). However, the start codon is not determined in ND1 in *R. longicrus* and *R. zhenhaiensis*. Four types of stop codons were complete stop codons (TAA, TAG, AGA, and AGG), and the other type was incomplete stop codons (T--) (Figure 3). The incomplete stop codons were the most common stop codon, which was used by 10 genes (ND1, ND2, COII, ATP8, ATP6, COIII, ND3, ND4, ND5, Cyt *b*).

Comparison of PCGs provide a better understanding of the evolutionary pattern of molecular evolution. The variable positions in each gene ranged from 35.50% (COIII) to 54.09% (ATP8), and the parsimony informative sites ranged from 30.27% (COIII) to 47.41% (ND5). The average uncorrected pairwise distances showed the heterogeneity of the evolutionary rate in each gene. The COI (0.02), COII (0.05), and COIII (0.05) had slowly evolutionary rate, whereas the gene of ATP8 (0.21) had the fastest rate (Table 3).

To further understand the evolutionary patterns among the 13 protein-coding genes, Ka/Ks was calculated for each protein-coding gene, respectively. The Ka/Ks ratio values of 13 PCGs were <1, which suggested purifying selection of these functional genes. The gene of COI (0.02) had the slowest evolution rates, and was then followed by COIII (0.04) and COII (0.04). The gene of ATP8 (0.18) had the fastest evolutionary rates and was then followed by ND3 (0.17), ND4L (0.15), and ND6 (0.14) (Table 3).

### 3.3. Transfer RNA, Ribosomal RNA Genes and Control Region

The mitogenome of *R. longicrus* and *R. zhenhaiensis* includes 22 tRNA genes, with the length ranging from 65 to 73 bp. The majority of tRNAs exhibit a secondary structure, usually a cloverleaf shape, with alternating double-helix stems and single-stranded nucleotides. However, tRNA^Ser(AGY)^ absented the dihydrouridine (DHU) arm.

Two ribosomal RNAs (12S rRNA and 16S rRNA) were identified on the H-strand. These two genes were separated by the gene of tRNA^Val^. The length of 12S rRNA and 16S rRNA were 931 bp (A+T = 53.38%) and 1,577bp (A+T = 57.36%) for *R. longicrus*, 931bp (A+T = 53.71%) and 1,576bp (A+T = 57.48%) for *R. zhenhaiensis*.

The control region is an important non-coding region in the mitogenome. It is located between Cyt *b* and tRNA^Leu(CUN)^, which spans 1,879 bp and 2,346 bp with A + T content of 61.6% and 59.3%, respectively. In *R. zhenhaiensis*, only one tandem repeats sequence of motif-1: 5′-TATGTTTAATAATCATTAATCTATCTGGATACTATCTC-3′ (38 nucleotides with 5 tandem repeats) was found. In *R. longicrus*, two tandem repeats were found, the motif-1 had 38 nucleotides with 4 tandem repeats, and the motif-2: 5′-TATGTTTAATAATCATTAACCTATCTAAGTACTATACCTA-3′ had 40 nucleotides with 2 tandem repeats).

### 3.4. Phylogenetic Relationships

We conducted a phylogenetic analysis of the mitogenomes of *Rana* species by using BI methods, estimating the phylogenetic tree using nucleotide sequences of 13 PCGs, 12S and 16S. All of the nodes exhibited high posterior probability values (Figure 4). In the phylogenetic tree, there were two major clades among *Rana*. The Clade A was consisted of three species, with *R. sylvatica* forms the sister group of two frogs (*R. catesbeiana* and *R. okaloosae*). The Clade B was consisted of the remaining species. The species of *R. longicrus* and *R. zhenhaiensis* formed a clade, which was a sister group to *R. omeimontis* and *R. chaochiaoensis*. Our analyses resolved two geographic lineages of *Rana*: Clade A (North America) was the sister group of Clade B (East Asia). The phylogeny of *Rana* reconstructed in our study was very similar to previous studies [2,44]. The main conflict involved the relationships among three species (*R.chensinensis*, *R. kukunoris* and *R. huanrensis*). Yuan et al. (2016) revealed the relationship of ((*R.chensinensis, R. huanrensis*)*, R. kukunoris*) with relative low nodal support using sequences of six nuclear and three mitochondrial loci (total of 7,250 bp) [2]. However, this study supported the relationship of ((*R.chensinensis, R. kukunoris*)*, R. huanrensis*) with high nodal support value, which was in agreement with the result of Yang et al. (2018) using two rRNAs and 13 PCGs [14]. Due to limited data on Europe and Central Asia groups in this study, we cannot provide more effective results to resolve detailed phylogenetic problems. The newly sequenced mitogenomes will provide useful information for better understanding the evolutionary history of genus *Rana* and will contribute to further taxonomic research within Ranidae.

### 3.5. Divergence-Time Estimation

The high ESS (effective sample size) value (>200) was identified for all parameters in the BEAST analysis to estimate the divergence time of *Rana* species (Figure 5). Dating analyses suggests that the most recent common ancestor (TMRCA) of *Rana* dates back to at 28.17 million years ago (Mya) [mean value; 95% of the highest posterior density (HPD), 23.07–34.69 Mya] (95% HPD, Mya) and dated to Oligocene (Figure 5), roughly consistent with the previous study [2]. It may be affected by the collision of India with Eurasia between 35–20 Ma [38]. TMRCA of Clade A and B were estimated at 22.00 Mya (95% HPD, 13.71–30.43 Mya) and 23.16 Mya (95% HPD, 19.19–328.11 Mya), respectively. Since 16.09 Mya, the diversification rate within Clade B began to increase during the Miocene (Figure 5), which indicated that the ancestral *Rana* fast dispersal across East Asia [38]. This study inferred the separation of *R. dybowskii* and *R. uenoi* dated at 8.13 Mya (95% HPD, 5.41–11.11 Mya) during the Middle and Late Miocene, which was consistent with the previous study [13], and was similar to the species diversification pattern in *Hyla* [45]. The Early Miocene diversification of *R. zhenhaiensis* and *R. longicrus* was formed by volcanism in Sikhote-Alin and Sakhalin in mid-Miocene-Pliocen [46]. We suggest that climate fluctuation and environmental change may have played an essential role in the species diversification in *Rana*.

### 3.6. Mitogenomic Rearrangements

Based on the comparison of genome organization, we detected three kind of gene arrangements. Our results showed that Pattern 1 was the most common type in *Rana* mitogenomes (Figure 6). Reconstructions of mitogenomic rearrangements pattern indicated that the ancestor of *Rana* consisted of Pattern 1. The *R. kunyuensis* and *R. amurensis* species have the same rearrangement, indicating that they have close relationships. The results showed that the mitochondrial gene arrangement was not related to species diversification, and several independent shifts happened in evolutionary history.

All rearrangements occurred between ND4 and tRNA^Thr^ region (Figure 7). Our results showed that Pattern 1 (the typical Neobatrachian-type arrangement) was the most common type, and Pattern 2 was shared by *R. kunyuensis* and *R. amurensis*. Compared with Pattern 1, one additional D-loop region was inserted into the upstream of tRNA^Thr^, and the ND5 was translocated from the typical tRNA^Ser(AGY)^ downstream to the tRNA^LeuCUN^ downstream. Pattern 3 was only discovered in *R. pyrenaica*, in which tRNA^Pro^-tRNA^Phe^ was not located between tRNA^Thr^ and 12S. 

The gene arrangement of vertebrate mitogenome is usually conservative, and gene recombination is relatively rare or random [49]. However, in Neobatrachian, tRNA genes were translocated from typical positions, and these tRNA genes formed a cluster upstream of the 12S rRNA gene [50]. Currently, the duplication and random loss (TDRL) model can be used to explain most of the animal mitogenome reorganization [51]. In this model, a part of the entire genome was duplicated accidentally due to replication errors (either slipped strand mispairing or inaccurate termination). Then one of the duplicates (or CR) was converted into a pseudogene and subsequently excised from the genome through the accumulation of natural mutations [52]. In this study, *R. longicrus* and *R. zhenhaiensis* mitogenome possessed the same gene order as Neobatrachian. Nine genes (between tRNA^Leu (CUN)^ and tRNA^Phe^) were tandem duplicated, forming two identical gene clusters. Then, after the random loss of nine genes, the gene cluster produced by tandem duplication formed the currently known type of mitochondrial gene rearrangement Figure 7b. This explanation was confirmed by previous studies [53,54].

Various types of mitogenome recombination have occurred in Ranidae species, which is also reflected in *Rana* genus [49]. In this study, we found a total of three kind of gene arrangement patterns in *Rana* species, which was consistent with the various types of mitogenome recombination in Ranidae species [55]. The mitogenome of *R. amurensis* and *R. kunyuensis* possessed two duplicate D-loop regions and this phenomenon was also found in other frogs [53]. The duplicated D-loop regions were similar to the original D-loop structure, which will result from homologous recombination between paralogous D-loop regions [44]. In general, the observed pattern is largely consistent with the TDRL model, the most general explanation for mitogenomic reorganization [56]. Subsequently, gene transfers via retrotransposition may be a pattern of gene rearrangement in animal mtDNAs [55]. We found mitogenome rearrangement in genus *Rana*, but it remains to be seen whether similar or new rearrangement patterns will occur in un-sequenced *Rana*.

## 4. Conclusions

In this study, we sequenced and annotated the mitogenome of *R. longicrus* and *R. zhenhaiensis* using the Illumina Novaseq 6000 platform for PE 2 × 150 bp sequencing. Circular mitogenomes of *Rana* displayed high size variation, with a mean length of 18,534 bp, ranging from 16,961 to 22,255 bp. The length of the mitogenome of *R. longicrus* was smaller than that of most *Rana* species. Differences in the length of the complete mitogenome are mainly caused by the control region. The mitogenome encoded a control region and a typical set of 37 genes containing 2 rRNA genes, 13 protein-coding genes, and 22 tRNA genes. The genome composition and characteristics were analyzed, RSCU and AT-skew values of the PCGs were calculated. In general, negative AT-skew and negative GC-skew are found in *Rana* mitogenomes, implying the specific bias toward T and C in nucleotide composition. The average uncorrected pairwise distances showed that the COI may be the most conserved protein coding gene, and ATP8 was the least conserved. The most common start codon was ATG and the incomplete stop codon was most common as a stop codon. To verify the gene order of mitogenome, multiple sequence alignment of the *Rana* mitogenome sequences was performed. There were three kinds of gene arrangement patterns in *Rana*. The gene recombination mainly occurred at the control region, gene of ND5 and tRNA. The results showed that the Pattern 1 was the most common type in *Rana* mitogenomes, the mitochondrial gene arrangement was not related to species diversification, and several independent shifts happened in evolutionary history. Climate fluctuation and environmental change may have played an essential role in the species diversification in *Rana*. The molecular data obtained in this study are valuable for research on the taxonomy, population genetics, and evolution of frogs in the genera *Rana*. In the future, sequencing more mitogenomes from various taxonomic levels will provide useful information for better understanding the evolutionary history and will contribute to further taxonomic research within Ranidae.

## Figures and Tables

**Figure 1 animals-12-01250-f001:**
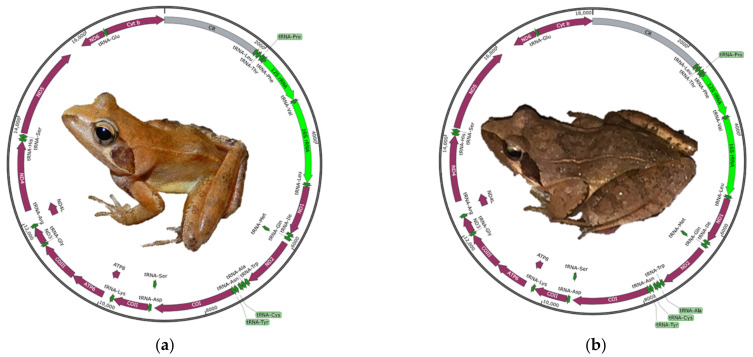
Map of the mitogenome of (**a**) *R. longicrus* and (**b**) *R. zhenhaiensis*. Arrows indicate the orientation of gene transcription. PCGs are shown as purple arrows, rRNA genes as green arrows, tRNA genes as dark green arrows and control region as gray arrows. Ticks in the inner cycle indicate the sequence length.

**Figure 2 animals-12-01250-f002:**
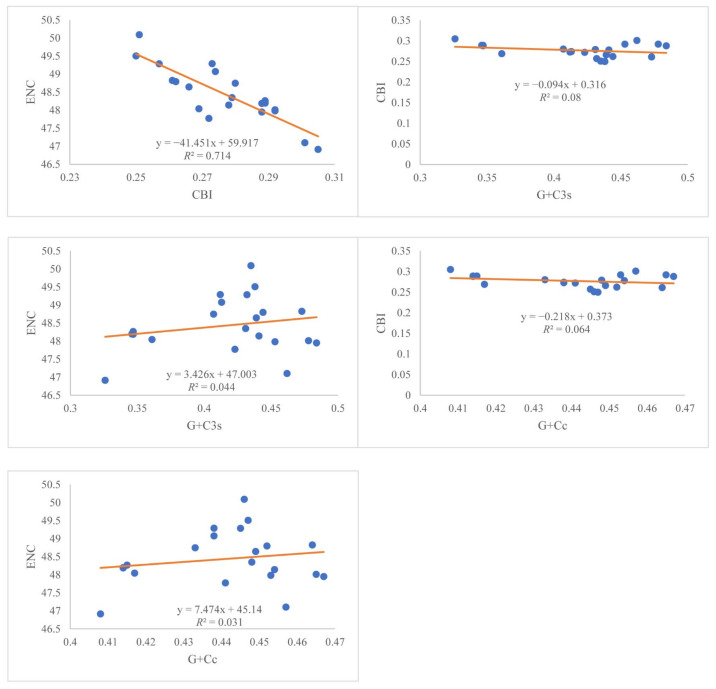
Evaluation of codon bias in mitogenomes of 17 Species of *Rana*. ENC, effective codon number. CBI, codon bias index. G + Cc, the content of G + C at all locations of the codon. G + C3s, G + C content of the third codon position.

**Figure 3 animals-12-01250-f003:**
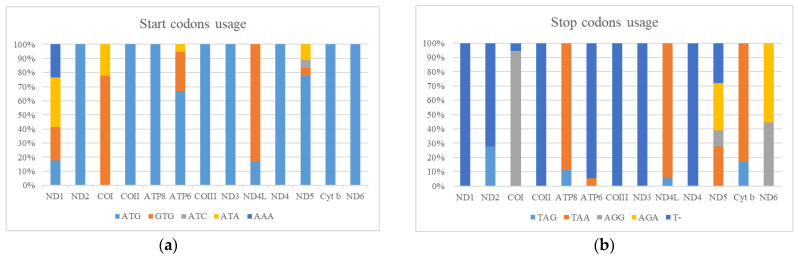
The usage of start (**a**) and stop (**b**) codons in the 13 mitochondrial protein-coding genes of *Rana*.

**Figure 4 animals-12-01250-f004:**
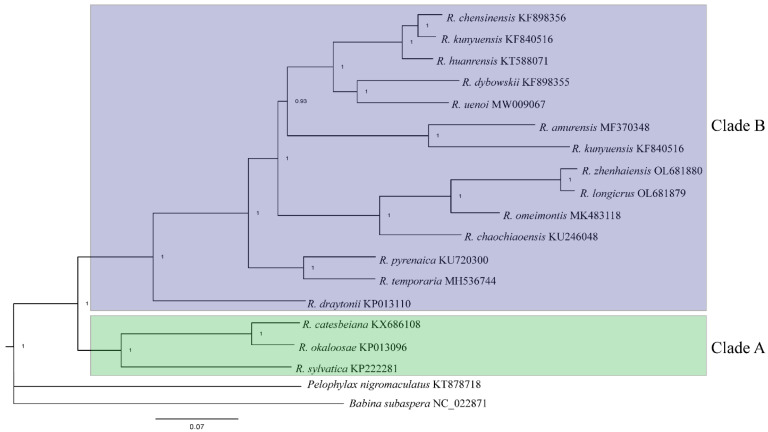
Phylogenetic tree using Bayesian inferences (BI) analyses based on mitochondrial genes (concatenated by 13 PCGs and 2 rRNAs). Posterior probabilities right to the nodes.

**Figure 5 animals-12-01250-f005:**
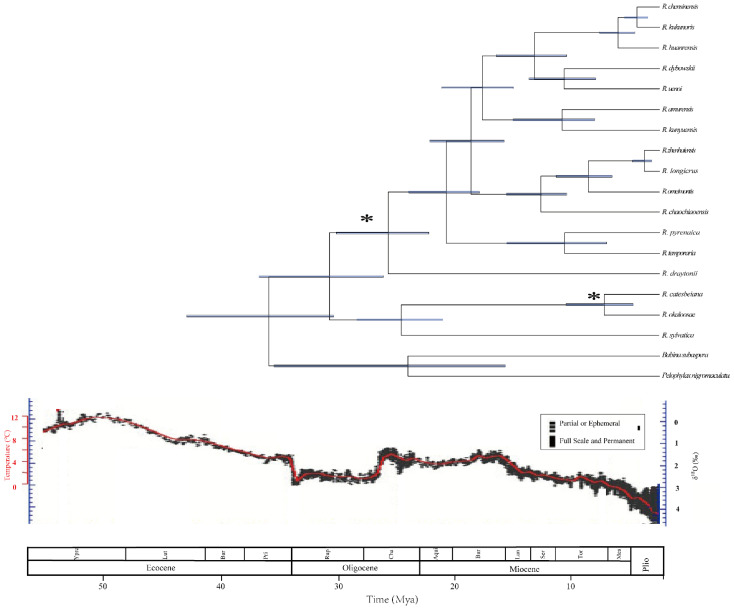
Time-calibrated phylogeny of *Rana* inferred from the complete mitogenomes. The symbol * indicates the calibration points. Horizontal blue bars represent 95% posterior density of the age on the node. The bottom axis is in million years ago (Mya). The climatic sequence of events including a global average δ^18^O curve (right-hand axis) derived from benthic foraminifera which mirrors the major global temperature trends [47,48].

**Figure 6 animals-12-01250-f006:**
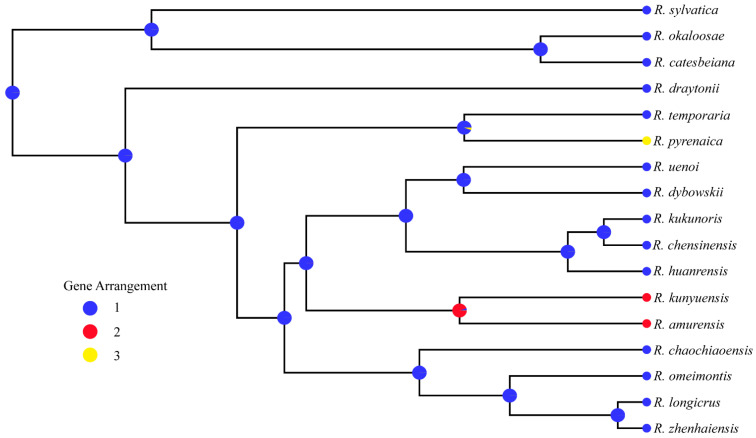
Reconstruction of ancestral mitogenomic rearrangement of *Rana*. Pie diagrams on nodes indicate the relative probability for each pattern.

**Figure 7 animals-12-01250-f007:**
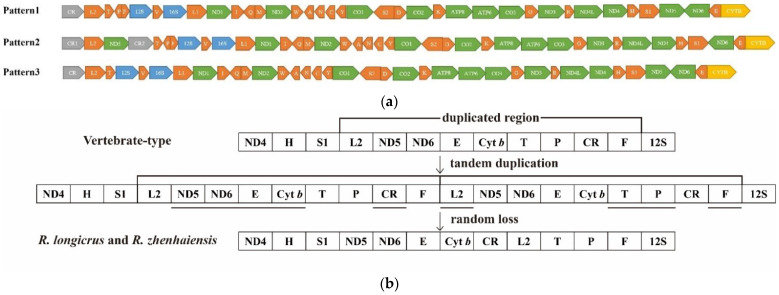
(**a**). Comparison of mitogenome arrangement of *Rana*. tRNAs are named using single-letter amino acid abbreviations. L1, L2, S1, and S2 represent tRNAs of Leu^(UUR)^, Leu^(CUN)^, Ser^(AGY)^, and Ser^(UCN)^, respectively. (**b**). A model for gene reorganization in the mtDNA genomes of *R. longicrus and R. zhenhaiensis*. After tandem duplication of a gene region is produced, multiple deletions of redundant genes occur. The gene order of *R. longicrus and R. zhenhaiensis* is the same as neobatrachians’ general one.

**Table 1 animals-12-01250-t001:** Species used in this study.

Species	Abbreviations	Length(bp)	GenBank
*Rana zhenhaiensis* *	zhe	18,006	OL681880
*Rana longicrus* *	lon	17,502	OL681879
*Rana huanrensis*	hua	19,253	KT588071
*Rana amurensis*	amu01	20,571	MF370348
*Rana amurensis*	amu02	18,470	KU343216
*Rana kunyuensis*	kun	22,255	KF840516
*Rana chensinensis*	che	18,808	KF898356
*Rana dybowskii*	dyb	18,864	KF898355
*Rana omeimontis*	ome	20,120	MK483118
*Rana kukunoris*	kuk	18,863	KU246049
*Rana chaochiaoensis*	cha	18,591	KU246048
*Rana draytonii*	dra	17,805	KP013110
*Rana catesbeiana*	cat01	17,681	KX686108
*Rana catesbeiana*	cat02	17,682	AB761267
*Rana catesbeiana*	cat03	18,241	KF049927
*Rana sylvatica*	syl	17,343	KP222281
*Rana okaloosae*	oka	17,504	KP013096
*Rana pyrenaica*	pyr	17,211	KU720300
*Rana temporaria*	tem	16,061	MH536744
*Rana uenoi*	uen	17,370	MW009067
*Babina subaspera*	Bsub	18,525	NC_022871
*Pelophylax nigromaculatus*	Pnig	17,567	KT878718

Sequenced mitogenomes in this study are noted by “*”.

**Table 2 animals-12-01250-t002:** The usage of amino acid coding codon.

AA	Codon	Count	Percentage (%)	RSCU	AA	Codon	Count	Percentage (%)	RSCU
Phe (F)	UUU	133.9	3.59	1.02	Ala (A)	GCA	69.9	1.87	0.87
Phe (F)	UUC	128.5	3.44	0.98	Ala (A)	GCG	13.4	0.36	0.17
Leu2 (L2)	UUA	119.8	3.21	1.15	Tyr (Y)	UAU	55.8	1.49	0.98
Leu2 (L2)	UUG	25.6	0.69	0.25	Tyr (Y)	UAC	58.4	1.56	1.02
Leu1 (L1)	CUU	135.2	3.62	1.3	His (H)	CAU	31.6	0.85	0.64
Leu1 (L1)	CUC	158.6	4.25	1.52	His (H)	CAC	66.9	1.79	1.36
Leu1 (L1)	CUA	143.8	3.85	1.38	Gln (Q)	CAA	75.5	2.02	1.78
Leu1 (L1)	CUG	43.4	1.16	0.42	Gln (Q)	CAG	9.3	0.25	0.22
Ile (I)	AUU	170.3	4.56	1.12	Asn (N)	AAU	60.1	1.61	0.94
Ile (I)	AUC	134.5	3.60	0.88	Asn (N)	AAC	67.6	1.81	1.06
Met (M)	AUA	115.2	3.09	1.42	Lys (K)	AAA	70.7	1.89	1.74
Met (M)	AUG	47	1.26	0.58	Lys (K)	AAG	10.8	0.29	0.26
Val (V)	GUU	57.4	1.54	1.17	Asp (D)	GAU	26.4	0.71	0.72
Val (V)	GUC	52.5	1.41	1.07	Asp (D)	GAC	46.6	1.25	1.28
Val (V)	GUA	61.5	1.65	1.25	Glu (E)	GAA	68.6	1.84	1.56
Val (V)	GUG	25.4	0.68	0.52	Glu (E)	GAG	19.5	0.52	0.44
Ser2 (S2)	UCU	63.2	1.69	1.43	Cys (C)	UGU	12.8	0.34	0.82
Ser2 (S2)	UCC	75.9	2.03	1.71	Cys (C)	UGC	18.3	0.49	1.18
Ser2 (S2)	UCA	66.6	1.78	1.5	Trp (W)	UGA	87.8	2.35	1.63
Ser2 (S2)	UCG	9.1	0.24	0.21	Trp (W)	UGG	19.8	0.53	0.37
Pro (P)	CCU	44.7	1.20	0.88	Arg (R)	CGU	12.1	0.32	0.65
Pro (P)	CCC	88.2	2.36	1.73	Arg (R)	CGC	22.6	0.61	1.23
Pro (P)	CCA	59.6	1.60	1.17	Arg (R)	CGA	32.8	0.88	1.78
Pro (P)	CCG	11.1	0.30	0.22	Arg (R)	CGG	6.2	0.17	0.34
Thr (T)	ACU	72.4	1.94	1.01	Ser (S1)	AGU	16.4	0.44	0.37
Thr (T)	ACC	102.2	2.74	1.43	Ser (S1)	AGC	34.6	0.93	0.78
Thr (T)	ACA	98.8	2.65	1.38	Gly (G)	GGU	28.8	0.77	0.52
Thr (T)	ACG	13.1	0.35	0.18	Gly (G)	GGC	81.5	2.18	1.47
Ala (A)	GCU	86.8	2.33	1.08	Gly (G)	GGA	64.7	1.73	1.17
Ala (A)	GCC	152.6	4.09	1.89	Gly (G)	GGG	46.9	1.26	0.85

**Table 3 animals-12-01250-t003:** Variation analysis and base composition of 13 PCG genes in *Rana*.

Gene	Length (bp)	%Vs	%Pis	%S	ts/tv	Ks	Ka	Ka/Ks	AT%	AT- Skew	GC- Skew	Aupd
ND1	960	48.02	41.67	6.35	4.26	0.98	0.08	0.08	56.24	−0.14	−0.41	0.12
ND2	1032	48.45	40.79	7.66	4.19	0.85	0.08	0.09	57.2	−0.03	−0.47	0.13
COI	1551	36.56	32.17	4.38	5.16	0.93	0.02	0.02	54.83	−0.09	−0.21	0.02
COII	687	35.66	30.71	4.95	5.99	0.78	0.03	0.04	56.53	0.04	−0.28	0.05
ATP8	159	54.09	42.14	11.95	5.79	0.55	0.1	0.18	61.89	0.04	−0.47	0.21
ATP6	693	45.74	38.67	7.07	5.27	0.96	0.06	0.06	57.35	−0.11	−0.47	0.09
COIII	783	35.50	30.27	5.24	5.35	0.8	0.03	0.04	53.58	−0.14	−0.28	0.05
ND3	339	50.44	44.84	5.60	4.26	0.53	0.09	0.17	54.87	−0.22	−0.36	0.15
ND4L	282	50.00	46.43	8.51	4.50	0.53	0.08	0.15	55.84	−0.11	−0.40	0.12
ND4	1359	44.81	37.97	6.84	4.31	0.92	0.06	0.07	56.81	−0.08	−0.41	0.09
ND5	1740	57.99	47.41	10.57	3.46	1.08	0.14	0.13	56.71	−0.09	−0.34	0.20
Cyt *b*	1140	39.56	33.51	6.05	4.18	1.00	0.03	0.03	54.00	−0.10	−0.4	0.05
ND6	486	50.41	43.62	6.80	3.98	0.84	0.12	0.14	52.81	−0.31	0.54	0.16

Vs: variable sites, Pis: parsimony informative sites, S: singleton, ts/tv: the estimated transition/transversion bias, AT-skew = (A − T)/(A + T), GC-skew = (G − C)/(G + C), Aupd: The average uncorrected pairwise distances.

## Data Availability

The genome sequence data that support the findings of this study are openly available in GenBank of NCBI at https://www.ncbi.nlm.nih.gov/ under the accession no. OL681879 and OL681880. The raw sequencing data have been deposited in GenBank associated BioProject, SRA, and Bio-Sample numbers are PRJNA785017, SUB10733125, SAMN23525929, and PRJNA784715, SUB10728085, SAMN23498843, respectively.

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
