# Peer review of "Comparative Mitogenomics of True Frogs (Ranidae, Anura), and Its Implications for the Phylogeny and Evolutionary History of Rana"

_animals, 2022, doi:10.3390/ani12101250_

Round 1

Reviewer 1 Report

This study mapped the mitogenome of 2 species in the genus Rana and compared results to 18 other Rana species and 2 outgroups.

They found that Rana showed 2 clades and four gene arrangement patterns. I found the manuscript complete, detailed and understandable.

Minor sentence structure fixes are necessary. For example, the sentence in the abstract "However, the evolutionary history and population genetics in Rana were not well understand, which may be limited by the absence of mitogenome information." could be corrected to However, the evolutionary history of the genus Rana is not well understood and may be limited by the absence of mitogenome information.

I recommend removing the phrase population genetics because I did not see reference to the topic in the main text.

Author Response

Reviewer 1

Comments and Suggestions for Authors

This study mapped the mitogenome of 2 species in the genus Rana and compared results to 18 other Rana species and 2 outgroups. They found that Rana showed 2 clades and four gene arrangement patterns. I found the manuscript complete, detailed and understandable.

>> Thank you for your constructive comments which are very helpful in preparing a revised manuscript. And many thanks for your encouragement.

Minor sentence structure fixes are necessary. For example, the sentence in the abstract "However, the evolutionary history and population genetics in Rana were not well understand, which may be limited by the absence of mitogenome information." could be corrected to However, the evolutionary history of the genus Rana is not well understood and may be limited by the absence of mitogenome information.

>>Revised. See in Line 32.

I recommend removing the phrase population genetics because I did not see reference to the topic in the main text.

>>Revised. See in Line 20 and Line 32.

Reviewer 2 Report

This manuscript reports sequencing of mitochondrial genomes of two frog species with comparative analysis from 18 species of genus Rana. Authors have analyzed different aspects such as mitochondrial genes features, phylogenetic relationships, and gene rearrangement mechanisms. The main findings of this study include the four kinds of gene arrangement patterns and the purify selection of mitochondrial genes in frog species. The mitogenomic analysis is adequate to support the conclusion of this work and overall manuscript is well written. However, I have the following two major concerns to assess the novelty of this work.

  1. The complete mitochondrial genome of frog Rana zhenhaiensis was already published elsewhere (see et al. Huang et al. 2017; GenBank ID: MN218687.1). However, in this study, it is claimed as newly sequenced mitogenome.
  2. After checking the given accession IDs from the “Data Availability Statement”, there is no record available in NCBI for any of these accession in both SRA and Genebank databases. The availability of this data is very important to evaluate the quality of mitogenome assemblies and annotations.

Many readers will also be very interested in this data for future work related to mitochondrial phylogenetics.

Minor comments:

  1. In the Line. 102. “A total of 6.0 Gb raw reads were generated by next-generation sequencing on the 102 Illumina platform”. It is not clear whether how many of raw reads correspond to each species? Further the specific Illumina platform (Miseq, Hiseq or other) used should be informed. Also, the sequence coverage/depth of this generated data should be informed.
  2. In the Line 133. “The tandem repeats were searched in the CR using the Tandem Repeats Finder program (https://tandem.bu.edu/trf/trf.html)”. This information is missing in the results sections how many and what type of repeats were found?
  3. It is recommended to change the term “Genomic Rearrangements” (Line. 225) by “Mitogenomic Rearrangements”. Genomic Rearrangements is actually used for whole genome (or chromosomes).
  4. The Figure 4, 5 and 6 represent and describe the data for the related topic of Genes organizations. It would be better to merge these as Panels (a), (b) and (c) into a single figure.

Author Response

Reviewer 2

Comments and Suggestions for Authors

This manuscript reports sequencing of mitochondrial genomes of two frog species with comparative analysis from 18 species of genus Rana. Authors have analyzed different aspects such as mitochondrial genes features, phylogenetic relationships, and gene rearrangement mechanisms. The main findings of this study include the four kinds of gene arrangement patterns and the purify selection of mitochondrial genes in frog species. The mitogenomic analysis is adequate to support the conclusion of this work and overall manuscript is well written. However, I have the following two major concerns to assess the novelty of this work.

>> Thank you for your constructive comments which are very helpful in preparing a revised manuscript. 

  1. The complete mitochondrial genome of frog Rana zhenhaiensis was already publishedelsewhere (see et al. Huang et al. 2017; GenBank ID: MN218687.1). However, in this study, it is claimed as newly sequenced mitogenome.

>> Thanks a lot. We have revised. Please see Line 77.

  1. After checking the given accession IDs from the “Data Availability Statement”, there is no record available in NCBI for any of these accession in both SRA and Genebank databases. The availability of this data is very important to evaluate the quality of mitogenome assemblies and annotations. Many readers will also be very interested in this data for future work related to mitochondrial phylogenetics.

>> Thank you for your careful reading. The data have been uploaded in GenBank, and it will be automatically released on December 1, 2022, or when the article is published. The details are shown in the attached file

Minor comments:

  1. In the Line. 102. “A total of 6.0 Gb raw reads were generated by next-generation sequencing on the 102 Illumina platform”. It is not clear whether how many of raw reads correspond to each species? Further the specific Illumina platform (Miseq, Hiseq or other) used should be informed. Also, the sequence coverage/depth of this generated data should be informed.

>> Revised. See in line 102, 113-114.

  1. In the Line 133. “The tandem repeats were searched in the CR using the Tandem Repeats Finder program (https://tandem.bu.edu/trf/trf.html)”. This information is missing in the results sections how many and what type of repeats were found?

>> Revised. See in Line 227-231.

  1. It is recommended to change the term “Genomic Rearrangements” (Line. 225) by “Mitogenomic Rearrangements”. Genomic Rearrangements is actually used for whole genome (or chromosomes).

>> Revised. See in Line 233.

  1. The Figure 4, 5 and 6 represent and describe the data for the related topic of Genes organizations. It would be better to merge these as Panels (a), (b) and (c) into a single figure.

>> Revised. See in Line 272-285.

Reviewer 3 Report

The manuscript entitled “The evolution of mitochondrial genomes in true frogs (Ranidae, Anura) revealed diverse gene arrangement in Rana species” address the mitogenome of two Rana sp. species of China. Although the data obtained is always useful for further studies, the novelty of the work is low, as the main contribution to the science is the data generated and not the conclusions we can extract from the work. However the work is scientifically correct, well written and in my opinion rigorous.   I think it could be interesting for a local audience, and as a basis for further evolutionary or ecological works. However, this work would benefit from an expansion, including echological analyses for both species, and relating it with mitogenome results per example. This would allow to reach more conclusions with a wider interest.

For this reason I propose the work for major revisions, recognizing that with piece of additional work it is publishable in Animals.

Author Response

Reviewer 3

Comments and Suggestions for Authors

The manuscript entitled “The evolution of mitochondrial genomes in true frogs (Ranidae, Anura) revealed diverse gene arrangement in Rana species” address the mitogenome of two Rana sp. species of China. Although the data obtained is always useful for further studies, the novelty of the work is low, as the main contribution to the science is the data generated and not the conclusions we can extract from the work. However the work is scientifically correct, well written and in my opinion rigorous.  I think it could be interesting for a local audience, and as a basis for further evolutionary or ecological works. However, this work would benefit from an expansion, including echological analyses for both species, and relating it with mitogenome results per example. This would allow to reach more conclusions with a wider interest. 

For this reason I propose the work for major revisions, recognizing that with piece of additional work it is publishable in Animals.

>> Thank you for your constructive comments. We read and study the recent mitogenome research papers published in Animals, then revised this manuscript. Two papers were cited in the references. See in Line 65-66.

Round 2

Reviewer 2 Report

I appreciate authors revisions. This revised manuscript is improved and I have one recommendation regarding figures. It is recommended to increase the text size in figure 1 and figure 4. I do not have any further comments.

Author Response

Reviewer 2

I appreciate authors revisions. This revised manuscript is improved and I have one recommendation regarding figures. It is recommended to increase the text size in figure 1 and figure 4. I do not have any further comments.

>> Done. Thank you for your comments.

Reviewer 3 Report

I still think this work needs additional work, with an extensive review of Rana sp. mithogenome. Authors have limited their review to only to references in Annimals and it is not enough.  They shoud inclede all knowledge available in all scientific literature. Thus, I can't recommend this work for publication in the present form. 

Author Response

Reviewer 3

I still think this work needs additional work, with an extensive review of Rana sp. mitogenome. Authors have limited their review to only to references in Animals and it is not enough.  They shoud inclede all knowledge available in all scientific literature. Thus, I can't recommend this work for publication in the present form.

>> Thank you for your advice. We have read, study, and cite the recent mitogenome research in Rana from the journal of Molecular Phylogenetics and Evolution, PLoS ONE, ZooKeys, Asian Herpetological Research, and so on. Then revised this manuscript.  This study was only invested effort in generating new genomic resources, which is always a welcome thing to see and crucial to facilitate future research in various fields of Rana research such as phylogenetics, population, conservation, functional genetics etc. And we want to reveal the spatiotemporal diversification of the genus Rana by using three genes (Table 1). But we think that these contents are of little relevance to this study.

>> Thanks again for your constructive comments.

Table 1. GenBank numbers for samples of the genus Rana used.

ID

Species

16S

Cyt b

ND2

1

Rana wuyiensis sp. nov.

MZ337980

MZ355497

MZ355540

2

Rana zhengi

MZ337992

MZ355509

MZ355552

3

Rana sangzhiensis

MZ338002

MZ355517

MZ355559

4

Rana jonhsi

MZ338010

MZ355525

MZ355566

5

Rana weiningensis

MZ338023

MZ355538

MZ355576

6

Rana amurensis

KX269203

KX269349

KX269418

7

Rana areolata

AY779229

KX269300

KX269369

8

Rana arvalis

KX269197

KX269344

KX269413

9

Rana asiatica

KX269200

KX269346

KX269415

10

Rana berlandieri

AY779235

KX269301

KX269370

11

Rana boylii

KX269178

KX269299

KX269368

12

Rana cascadae

KX269176

KX269302

KX269371

13

Rana catesbeiana

KX269208

KX269354

KX269423

14

Rana chaochiaoensis

KX269192

KX269339

KX269408

15

Rana chensinensis

KX269186

KX269333

KX269402

16

Rana chiricahuensis

AY779225

KX269303

KX269372

17

Rana clamitans

AY779204

KX269304

KX269373

18

Rana coreana

KX269202

KX269348

KX269417

19

Rana culaiensis

KX269190

KX269337

KX269406

20

Rana dunni

AY779222

KX269305

KX269374

21

Rana dybowski

KX269188

KX269335

KX269404

22

Rana forreri

AY779233

GU184219

GU184250

23

Rana graeca

KX269199

KX269345

KX269414

24

Rana hanluica

KX269191

KX269338

KX269407

25

Rana huanrensis

KX269183

KX269330

KX269400

26

Rana iberica

KX269195

KX269342

KX269411

27

Rana japonica

KX269220

KX269364

KX269434

28

Rana kukunoris

KX269185

KX269332

KX269401

29

Rana kunyuensis

KX269201

KX269347

KX269416

30

Rana longicrus

KX269189

KX269336

KX269405

31

Rana luteiventris

KX269213

KX269358

KX269428

32

Rana macrocnemis

KX269194

KX269341

KX269410

33

Rana macroglossa

AY779243

KX269306

KX269376

34

Rana maculata

AY779207

KX269307

KX269377

35

Rana magnaocularis

AY779239

KX269308

KX269378

36

Rana montezumae

AY779223

KX269309

KX269379

37

Rana neovolcanica

AY779236

KX269310

KX269380

38

Rana omeimontis

KX269193

KX269340

KX269409

39

Rana omiltemana

AY779238

KX269311

KX269381

40

Rana ornativentris

KX269187

KX269334

KX269403

41

Rana palustris

KX269207

KX269353

KX269422

42

Rana psilonota

AY779217

KX269312

KX269382

43

Rana pustulosa

AY779220

KX269313

KX269383

44

Rana sakuraii

KX269205

KX269351

KX269420

45

Rana sauteri

KX269204

KX269350

KX269419

46

Rana septentrionalis

KX269179

KX269314

KX269384

47

Rana shuchinae

KX269210

KX269356

KX269425

48

Rana sierrae

KX269211

KX269357

KX269426

49

Rana sierramadrensis

AY779216

KX269315

KX269385

50

Rana spectabilis

AY779227

KX269320

KX269390

51

Rana sphenocephala

AY779251

KX269321

KX269391

……

……

Round 3

Reviewer 3 Report

Although I disagree with the authors (the contextualization of Rana sp. mithogenome is necessary in order to provide clear conclusions and not only data), I appreciate the effort to review additional references.

Author Response

>> Thank you for your encouragement and constructive comments. We have learned a lot from your advice, which will be of great benefit in our future study. And we have modified the grammar a in the manuscript to make the article more readable. Thanks again.